# Investigation of the Molecular Epidemiology and Evolution of Circulating Severe Acute Respiratory Syndrome Coronavirus 2 in Thailand from 2020 to 2022 via Next-Generation Sequencing

**DOI:** 10.3390/v15061394

**Published:** 2023-06-19

**Authors:** Jiratchaya Puenpa, Vorthon Sawaswong, Pattaraporn Nimsamer, Sunchai Payungporn, Patthaya Rattanakomol, Nutsada Saengdao, Jira Chansaenroj, Ritthideach Yorsaeng, Kamol Suwannakarn, Yong Poovorawan

**Affiliations:** 1Center of Excellence in Clinical Virology, Department of Pediatrics, Faculty of Medicine, Chulalongkorn University, Bangkok 10330, Thailand; jiratchaya.pu@gmail.com (J.P.); k_iis-z@hotmail.com (P.R.); job151@hotmail.com (J.C.); ritthideach.yor@gmail.com (R.Y.); 2Center of Excellence in Systems Microbiology, Department of Biochemistry, Faculty of Medicine, Chulalongkorn University, Bangkok 10330, Thailand; vorthon007.giftedcru@gmail.com (V.S.); k.knim@hotmail.com (P.N.); sp.medbiochemcu@gmail.com (S.P.); 3Department of Microbiology, Faculty of Medicine, Siriraj Hospital, Mahidol University, Bangkok 10700, Thailand; natsada.sae@student.mahidol.edu (N.S.); kamol.suw@mahidol.edu (K.S.); 4FRS(T), The Royal Society of Thailand, Sanam Sueapa, Dusit, Bangkok 10300, Thailand

**Keywords:** SARS-CoV-2, COVID-19, evolution, complete genome sequencing, molecular epidemiology, next-generation sequencing

## Abstract

Coronavirus disease 2019 (COVID-19) is an infectious condition caused by the severe acute respiratory syndrome coronavirus-2 (SARS-CoV-2), which surfaced in Thailand in early 2020. The current study investigated the SARS-CoV-2 lineages circulating in Thailand and their evolutionary history. Complete genome sequencing of 210 SARS-CoV-2 samples collected from collaborating hospitals and the Institute of Urban Disease Control and Prevention over two years, from December 2020 to July 2022, was performed using next-generation sequencing technology. Multiple lineage introductions were observed before the emergence of the B.1.1.529 omicron variant, including B.1.36.16, B.1.351, B.1.1, B.1.1.7, B.1.524, AY.30, and B.1.617.2. The B.1.1.529 omicron variant was subsequently detected between January 2022 and June 2022. The evolutionary rate for the spike gene of SARS-CoV-2 was estimated to be between 0.87 and 1.71 × 10^−3^ substitutions per site per year. There was a substantial prevalence of the predominant mutations C25672T (L94F), C25961T (T190I), and G26167T (V259L) in the ORF3a gene during the Thailand outbreaks. Complete genome sequencing can enhance the prediction of future variant changes in viral genomes, which is crucial to ensuring that vaccine strains are protective against worldwide outbreaks.

## 1. Introduction

Over the past few decades, RNA viruses belonging to the Coronaviridae family have cyclically caused life-threatening illnesses in the human population due to zoonotic spillover. In 2002, the severe acute respiratory syndrome coronavirus originated in Guangdong, China, and gave rise to a pandemic of atypical pneumonia, resulting in 8437 confirmed cases and 813 deaths [1]. In 2012, the Middle East respiratory syndrome coronavirus, first reported in Saudi Arabia, caused severe respiratory illness and death in 27 countries, with 858 confirmed fatalities [2]. More recently, severe acute respiratory syndrome coronavirus 2 (SARS-CoV-2) emerged in Wuhan, China, in late December 2019 and has been declared an etiological cause of the COVID-19 pandemic since March 2020 [3]. As of 31 May 2023, the number of confirmed COVID-19 cases was > 760 million worldwide, including almost 7 million fatal cases [4]. The COVID-19 pandemic has continuously and progressively imposed a severe burden on economic and public health systems.

SARS-CoV-2 is an enveloped, positive-sense single-stranded RNA virus with a large genome of approximately 27–32 kb [5]. In contrast to other RNA viruses, coronaviruses harbor the proofreading activity of nonstructural protein 14 (nsp14ExoN), promoting the replication fidelity of their RNA-dependent RNA polymerase (RdRp), leading to a low mutation rate [6,7]. Continuous large-scale circulation of SARS-CoV-2 and an inequitable distribution of vaccines and antiviral drugs have resulted in stochastic intra- and inter-transmission events at the population level, which have driven a degree of viral adaptation and escape from host immunity. To maintain essential genetic information while adapting its molecular ligand to fit the host milieu, genetic diversification of the virus mainly occurs in the spike (S) region, resulting in broadened tissue tropism and host range and waning host immune defense [8,9]. The emergence of new variants is characterized by alteration of the spike gene, and variants of concern (VOCs) are classified based on consequences, i.e., increased transmission, reduced vaccine and antiviral effectiveness, reduced treatment efficacy, and altered clinical disease presentation [10]. To date, the world has been confronted with five VOCs—alpha, beta, gamma, delta, and more recently, omicron—the common sublineages of which are BA.1 to BA.5. Over ten million SARS-CoV-2 genome data entries collected worldwide are available from the Global Initiative on Sharing All Influenza Data (GISAID; http://www.gisaid.org, accessed on 15 January 2023) [11]. The study of the evolution of the virus, alongside the investigation of host-viral interaction, is valuable for improving coping strategies and health policies and predicting the evolutionary trajectory of the virus.

As of May 24, 2023, the time of writing, there had been 4,738,988 confirmed COVID-19 cases in Thailand and a total of 34,053 reported deaths [12]. Since January 2020, Thailand has experienced five COVID-19 waves, and the omicron variant is dominating the fifth ongoing wave. Despite >25 million members of the population having had a third vaccine, COVID-19 remained a significant public health burden, affecting both fully vaccinated and vulnerable populations.

The present study investigated the molecular epidemiological trends and evolutionary history of SARS-CoV-2 in Thailand from December 2020 to July 2022. The genetic traits of Thai sequence variants and their phylogenetic relationships with other globally published variants were comprehensively analyzed via full-genome sequence comparisons.

## 2. Materials and Methods

### 2.1. Sample Collection and Processing

A total of 210 full-length SARS-CoV-2 genomes were successfully sequenced from individuals diagnosed with COVID-19 in Thailand from December 2020 to July 2022 (waves two to five). Of these, 63 sequences were detected before the B.1.1.529 omicron variant began to predominate. They were collected from various regions in Thailand, including Bangkok, Samut Sakhon, LobBuri, Narathiwat, and Yala, from December 2020 to December 2021. The remaining 147 sequences were collected during B.1.1.529 omicron variant predominance between January 2022 and July 2022 and were obtained exclusively from patients in Bangkok.

All nasopharyngeal samples included in the study were collected from collaborating hospitals and the Institute of Urban Disease Control and Prevention. These samples routinely tested positive for SARS-CoV-2 in multiplex real-time reverse transcription polymerase chain reaction (RT-PCR) assays, as described previously [13]. A magLEAD 12gC instrument (Precision System Science, Chiba, Japan) was used to extract nucleic acid from a 200-μL aliquot of supernatant in accordance with the manufacturer’s instructions, which was then analyzed at our laboratory.

### 2.2. Genomic Sequencing of SARS-CoV-2

To investigate the molecular epidemiology and evolution of SARS-CoV-2, further analysis of qRT-PCR-positive samples with CT values below 25 was conducted using next-generation sequencing (NGS). The Celemics comprehensive respiratory virus panel (Celemics Inc., Incheon, Republic of Korea) was used to sequence and identify complete SARS-CoV-2 genomes. Briefly, the RNA extraction process involved mixing 25 ng of extracted RNA with an RNA fragment buffer mix to facilitate fragmentation. First-strand cDNA was then synthesized using a first-strand synthesis master mix. The 1st-strand cDNA underwent double-stranded cDNA construction via incubation at 16 °C for 60 min with a 2nd-strand synthesis-1 mix, followed by a 2nd-strand synthesis-2 mix at 25 °C for 15 min. The double-stranded cDNA was cleaned, repaired, and added to poly(A) tail oligomers in a 5 ERA buffer mix. After multiple incubation steps at different temperatures, the A-tailed DNA was ligated with adaptors in a ligation reaction mix at 20 °C for 15 min. The ligated DNA was purified using CeleMag cleanup beads, amplified, and transformed into an adaptor-ligated library using CLM polymerase and UDI primers, in accordance with the manufacturer’s instructions. The constructed DNA library was assessed for quantity and quality via automated capillary gel electrophoresis (QIAxcel; Qiagen, Hilden, Germany) to ensure the presence of 200- to 400-bp DNA fragments. The DNA libraries were then subjected to NGS using the Illumina NextSeq 500 system with the mid/high-output kit v2.5 (300 cycles). The resulting FASTQ data were trimmed, assembled, and analyzed using the Celemics Virus Verifier pipeline, facilitating the identification and generation of consensus sequences for the SARS-CoV-2 genome. Any nucleotide gaps found in the assembled SARS-CoV-2 FASTQ sequences were filled by incorporating nucleotide sequences obtained from conventional RT-PCR-derived Sanger sequencing using primers specifically designed for those gaps.

### 2.3. Phylogenetic Analysis and Evolutionary Dynamics

The complete genome sequences acquired were compared with publicly available sequence data from GISAID. The sequence dataset was constructed with BioEdit v7.2.6 software [14] and aligned using CLUSTAL W at the European Bioinformatics Institute web server [15]. The diversity of SARS-CoV-2 lineages was analyzed with the maximum-likelihood phylogenetic method available in the MEGA program (v7) [16]. The Kimura two-parameter model with a gamma distribution (Γ) was selected as the substitution model in the analyses. The statistical consistency of tree nodes was determined via the bootstrap method (1000 random samplings).

A time-scaled phylogenetic tree for complete genome sequences was reconstructed with the BEAST version 1.10.4 program [17]. An uncorrelated lognormal prior distribution of nucleotide substitution rates among lineages and three independent Markov chain Monte Carlo (MCMC) procedures were used for Bayesian phylogenetic analyses. The general time-reversible model with a 4-category gamma-distributed rate variation across sites was used as the nucleotide substitution model. Bayesian Markov chain Monte Carlo analysis was run for 120 million steps and sampled every 300 steps from the posterior distribution. Tracer version 1.7.1 (http://tree.bio.ed.ac.uk/software/tracer/, accessed on 7 January 2023) was used to assess the convergence of all parameters (an adequate operator sample size of >200). The maximum clade credibility tree was summarized as maximum clade credibility (MCC) trees using the TreeAnnotator v1.10.4 tool (http://beast.bio.ed.ac.uk/treeannotator, accessed on 7 January 2023) after discarding the first 10% as burn-in, and then visualized in FigTree.

### 2.4. Nucleotide Sequence Accession IDs

Genome sequences generated in this study were deposited in the GISAID (https://www.gisaid.org, accessed on 10 March 2023) databases. Accession IDs are available in Appendix A.

## 3. Results

### 3.1. Divergence and Amino Acid Variations in SARS-CoV-2 Strains Detected before the Predominance of the B.1.1.529 Omicron Variant

The SARS-CoV-2 outbreaks in Thailand before the emergence of the B.1.1.529 omicron variant were classified into four waves. The first wave occurred from March 2020 to April 2020, the second from late December 2020 to January 2021, the third from April 2021 to July 2021, and the fourth from August 2021 to December 2021 [18]. A total of 63 sequences were collected prior to the predominance of the B.1.1.529 omicron variant. Of these, 14 sequences were sampled during the second wave, with 12 belonging to lineage B.1.36.16 and one each belonging to B.1.351 and B.1.1 lineages. There were 22 sequences from the third wave outbreak, with the alpha variant (19 sequences) being the most common, followed by lineage B.1.524 (3 sequences). During the fourth wave, 27 sequences were collected, with 17 belonging to lineage AY.30 and 10 belonging to lineage B.1.617.2.

Phylogenetic analysis revealed the dynamic nature of the epidemic in Thailand, and molecular changes in the SARS-CoV-2 genome were detected before the B.1.1.529 omicron variant predominated (Figure 1). Several disparate lineages were identified, with an initial lineage B (clade L) linked to early Bangkok cases dating from February 2019, including lineage A (clade S) and lineage B.1 (clades G, GH, and GR). All the lineages in the first epidemic wave except lineage B (clade L) probably emerged before April 2020. Lineage B.1.36.16 (clade GH) was found in July and August 2020 and was established near the beginning of the second epidemic wave. Most SARS-CoV-2 collected from the third epidemic wave belonged to lineage B.1.1.7 (clade GRY/alpha), with a few belonging to lineage B.1.524 (clade G). The third epidemic wave’s divergence time estimate for lineage B.1.1.7 (clade GRY/alpha) was December 2020. Phylogenetic analysis in the current study indicated that two lineages dominated the fourth epidemic wave: AY.30 (clade GK/delta) and B.1.617.2 (clade GK/delta). We estimated that interpersonal transmission of the fourth wave lineage began in January 2021. Its spread was sustained in April 2021 for lineage AY.30 (clade GK/delta) and in May 2021 for lineage B.1.617.2 (clade GK/delta).

Due to the error-prone nature of viral RNA genome replication, we analyzed crucial amino acid replacements in SARS-CoV-2 proteins from the samples acquired in this study from the second wave to the fourth epidemic wave. The 63 SARS-CoV-2 sequences identified in this study were combined with 67 published Thai samples to obtain a dataset of 130 sequences. The positions of amino acid substitutions in SARS-CoV-2 proteins and their relative frequencies in the entire set of 130 genomes were aligned and compared to the first isolate identified in December 2019, Wuhan-Hu-1 (Figure 2). Comparative analysis of the SARS-CoV-2 sequences revealed amino acid changes in all genome samples, most of which were scattered in nonstructural proteins. The nsp3, nsp14, and nsp2 viral proteins changed at 38, 22, and 17 amino acid positions, respectively. In S, N, and M, there were a total of 46, 24, and 10 amino acid position changes in the structural proteins, respectively. There were only three amino acid position changes in the E protein. There were minimal changes in nsp5, nsp7, nsp8, nsp9, nsp10, nsp16, and ORF6, and these changes were present in approximately 0.8–1.5% of the genome samples. Some amino acid substitutions in nsp3, nsp12, nsp13, M, S, ORF3a, ORF7a, ORF8, and N were present in >20% of the genomes sampled.

### 3.2. Evaluation of the Evolutionary History of SARS-CoV-2 in Thailand

To investigate the evolutionary history of SARS-CoV-2 outbreaks in Thailand, a phylogenetic analysis of the spike sequence samples obtained and the SARS-CoV-2 reference sequence Wuhan-Hu-1 (accession NC_045512) was conducted. Relationships between the Thailand SARS-CoV-2 variants and the dates of their emergence in Thailand are shown in Figure 3. The nucleotide substitution rate for the sampled population was estimated to be 1.24 × 10^−3^ (95% highest density interval 0.87–1.71 × 10^−3^) substitutions per site per year. The estimated time to the most recent common ancestor (tMRCA) of SARS-CoV-2 was 2.7 years for the most recent strain analyzed. The tMRCAs for the omicron sublineages BA.1 and BA.2 were approximately 0.8 and 0.6 years, respectively. The tMRCA for the omicron sublineages BA.4 and BA.5 was as recent as 0.2 years.

In Thailand, the Sinovac-CoronaVac vaccine was initially approved for use in late February 2021 (Figure 4). Following the outbreak of the third wave with the Alpha variant, the AstraZeneca vaccine and Sinopharm were administered to the Thai population in June 2021. During the fourth wave outbreak with the Delta variant, approximately 10% of the Thai population had received full vaccination, and the Pfizer-BioNTech vaccine was first used in Thailand in August 2021. The Thai population received the Moderna vaccine for the first time in November 2021. During the fifth wave outbreak with the Omicron variant, Thailand achieved a fully vaccinated rate of over 70%.

### 3.3. SARS-CoV-2 Omicron Sublineage BA.1 Genetic Characterization

The 63 Thailand B.1.1.529 omicron sequences identified in the present study were combined with 145 publicly available SARS-CoV-2 genome sequences identified worldwide, resulting in a comprehensive dataset of 208 sequences. All sequences in the present study in the B.1.1.529/BA.1 lineage were collected between January 2022 and May 2022. Analysis using clade-defining sequences (https://clades.nextstrain.org/, accessed on 7 January 2023) identified the Thailand sequences as 7 sublineages from the parent lineage B.1.1.529/BA.1 (Figure 5).

To investigate the mutation profile of the SARS-CoV-2 omicron BA.1 variant in the Thailand dataset, the 63 viral sequences identified in the current study and the dataset of 2951 viral sequences downloaded from the GISAID database were analyzed. The sequences were analyzed using the Nextclade Webtool to identify the most common mutations and characteristics of the Thailand dataset [19]. The majority of sequences (*n* = 44) were classified as sublineage BA.1.1 and shared the R346K (G22599A) substitution in the spike protein. Mutations in sublineage BA.1.1 (C2470T, C14805T, T19632C, and A26530G) were dominant, with a frequency > 50% in the Thailand genomes. With regard to the BA.1 variant from Thailand, T2019C (M585T in ORF1a), C2470T, G6850T, and G23628A (S689N in S) were present at frequencies > 10% (Table 1).

C14117T and A26530G were present at high frequencies (>60%) in Thai viral genomes in the sublineage BA.1.1.15. G2894A and G26167T mutations dominated (>10%) in Thailand’s genome sublineage BA.1.1.18. C4113T, C5672T, and A26530G mutations (>40%), followed by T851C, C10605T, C12084T, G15850A, and G28436T (<10%), were present in Thailand sublineages BA.1.17/BA.1.17.2. In this study, one Thailand strain (EPI_ISL_12176269) was identified as sublineage BA.1.22, and it was detected in March 2022. One Thailand strain contained six genetic variations: A3301T (ORF1a:L1012F), G11417T (ORF1a:V3718F), C15738T, C17285T (ORF1b:S1273L), C20719T, and C27494T (ORF7a:P34L). All strains in the sublineage BA.1.22 shared the unique mutations G3182A (ORF1a:E973K) and G5515T.

### 3.4. SARS-CoV-2 Omicron Sublineage BA.2 Genetic Characterization

BA.2 and its sublineages accounted for 29.5% of all variants among the sequenced samples. In this study, 62 omicron sublineage BA.2 variants obtained in Thailand were analyzed for the period from January 2022 to June 2022. In phylogenetic analysis, 53% (33/62) were classified as sublineage BA.2, 23% (14/62) as BA.2.10, 16% (10/62) as BA.2.9, 5% (3/62) as BA.2.27, and 3% (2/62) as BA.2.3 (Figure 6).

To further characterize the genomes of SARS-CoV-2 omicron BA.2 and its sublineages in the Thailand viral population, an analysis of sequence variants across the entire viral genome was conducted, comparing them to the Wuhan-Hu1 strain (MN908947). The mutations C241T, T22882G (S:N440K), and C23854A (S:N764K) were present at high frequencies (>80%) in the genomes of Thailand viral sublineage BA.2, followed by C7471T and C25416T (>40%) (Table 2). The BA.2.27 sublineage, primarily identified in Thailand, has been detected in several other countries, including the United Kingdom, France, the United States, and India. Within the BA.2.27 sublineage, mutations C241T and C10198T were present at frequencies > 80% in the 5′UTR and ORF1a regions, whereas mutations C17745T, C19610T (ORF1b:T2048I), C25672T (ORF3a:L94F), and G28739T (N:A156S) were present at frequencies < 10%. The BA.2.3 sublineage has also been identified in several Asian countries, primarily the Philippines, Japan, and South Korea. The BA.2.3 sublineage exhibited dominant mutations (C241T and A21222G) with frequencies > 75% in the Thai population. Mutations C832T and T7282C were also present at frequencies > 20%, and mutations in the ORF1b region were present at frequencies > 8%.

The BA.2.9 sublineage characterized by the H78Y mutation in ORF3a was predominantly circulating in Europe, with an exceptionally high prevalence observed in Denmark. That sublineage, which shares the V1393A mutation in ORF1a, was most commonly detected in Thailand but has also been identified in Germany, Israel, Japan, and Denmark. Among the BA.2.9/BA.2.9.5 sublineages, Thailand variants have frequencies > 3% and are located in ORF1a, ORF1b, ORF9b, and the spike protein. In the Thailand sublineage BA.2.10, the mutations T7813C and C25961T were present at frequencies > 10%.

### 3.5. SARS-CoV-2 Omicron Sublineage BA.4 and BA.5 Genetic Characterization

The omicron sublineages BA.4 and BA.5 comprised variants that were detected in Thailand during June 2022 and July 2022. A phylogenetic tree based on complete genome sequences was constructed to investigate genetic relationships between Thailand’s BA.4 and BA.5 variants and global BA.4 and BA.5 variants (Figure 7). The complete genomes of the Thailand BA.4 and BA.5 variants were compared with a set of 110 SARS-CoV-2 genomes publicly available from GISAID. In the tree, 7/210 sequences (3.3%) were categorized as sublineage BA.4, and 15/210 (7.1%) were categorized as sublineage BA.5. The Thailand BA.5 sequences were categorized into four subtypes: BA.5.2, BA.5.2.1, BA.5.2.22, and BA.5.2.26, as determined by an analysis using clade-defining sequences available at (https://clades.nextstrain.org/, accessed on 7 January 2023). In Thailand, BA.4 and its sublineages exhibited high frequencies (>70%) of C241T, G6680A (ORF1a:A2139T), A22786C (Spike:R408S), T22882G (Spike:N440K), and T24163C mutations (Table 3). The most frequent mutations (>80%) present in Thailand BA.5 and its sublineages were C16616A (ORF1b:T1050N), A18163G (ORF1b:I1566V), T22882G (Spike:N440K), C23854A (Spike:N764K), and C26270T (E:T9I).

## 4. Discussion

In this study, the genomic variation and molecular phylogeny of 210 SARS-CoV-2 strains identified in Thailand from December 2020 to July 2022 were characterized using complete genome sequences. Classification analysis identified 31 distinct SARS-CoV-2 lineages in the samples. Similar findings have also been reported in populations in Malaysia [20], Hong Kong [21], and India [22]. Among the 31 different lineages identified in this study, seven were detected before the emergence of the B.1.1.529 omicron variant, including B.1.36.16, B.1.351, B.1.1, B.1.1.7, B.1.524, AY.30, and B.1.617.2. Previous studies indicate that Thailand experienced its first COVID-19 wave between March 2020 and April 2020, during which the prevalent lineages identified were A, B, and B.1 [13]. Before the B.1.1.529 omicron variant became predominant the majority of lineages belonged to three groups; B.1.36.16 (second wave), alpha (third wave), and delta (fourth wave). On 24 November 2021 a woman who had traveled to Africa was recorded as the first occurrence of BA.1 in Thailand, with the GISAID identifier EPI_ISL_7398758. The prevalence of BA.1 peaked between January 2022 and February 2022, then it shifted to BA.2 in the following months [18]. The variants BA.4 and BA.5 were detected in our samples during June 2022 and July 2022. After Thai individuals received full vaccination coverage of over 70%, there has been a decrease in the number of SARS-CoV-2 infections.

Based on Bayesian analyses with the tip-dating method, the rate of evolutionary change in the SARS-CoV-2 spike region was 1.24 × 10^−3^ substitutions per site per year, which is concordant with previous studies [23,24,25,26]. This rate of change is comparable to that observed in other human coronaviruses [27], but it is nearly three times higher than the reported mutation rate of human influenza B [28]. A previous study examined the global evolution rate of SARS-CoV-2 during the early stages of the outbreak and reported an estimated mean nucleotide mutation rate ranging from 1.79 × 10^−3^ to 1.83 × 10^−3^ substitutions per site per year [29]. In another study, it was suggested that the incubation period, serial interval, and generation time of SARS-CoV-2 have progressively decreased with the emergence of each new VOC [30].

The SARS-CoV-2 genome has been undergoing rapid evolution throughout the pandemic, with evidence suggesting that mutations in the genome affect the virus’s virulence [31]. The current study identified distinctive genomic patterns of synonymous and missense variants linked to the distribution of lineages in Thailand. The genetic variations observed in the Thailand isolates predominantly occurred within nonstructural proteins. Previous studies have also reported similar findings [32]. The ORF3a gene encodes a protein crucial in modulating inflammation, antiviral responses, and apoptosis processes [33]. In the present study, a notable prevalence of the dominant mutations C25672T (L94F), C25961T (T190I), and G26167T (V259L) was observed within this gene in the Thailand isolates.

Although most structural proteins remained conserved, the spike protein exhibited multiple mutations, notably the dominant variant carrying the D614G mutation, which is frequently associated with enhanced viral infectivity [34]. The spike protein is widely recognized for facilitating infection via interaction with the angiotensin-converting enzyme 2 (ACE2) receptor on the surface of human host cells [35,36]. In the current study, the dominant mutations in the spike gene at R408S, N440K, and N764K were observed at a frequency of >70% in Thailand isolates. In previous studies, multiple mutations were identified in the receptor-binding domain of VOCs that enhanced ACE2 binding affinity and facilitated evasion of antibody binding [37,38]. For example, the K417N, L452R, E484K, F486V, and N501Y mutations, present in most VOCs, were also detected in the samples isolated in Thailand in the current study. The S689N mutation in the spike gene, which first appeared in unassigned variants in May 2020, was also detected in the BA.1 lineage. The S689N mutation was detected in multiple other variants, including B.1.1.7, B.1.258.11, B.1.351, and B.1.617.2.

The N protein maintains the genome structure inside the viral envelope and is also involved in viral assembly and budding [39,40]. Its high degree of conservation has led to its utilization for diagnostics and the investigation of it as a target for new vaccines [41,42]. R203K/G204R substitutions in the N protein have been linked to enhanced SARS-CoV-2 infectivity, fitness, and virulence [43,44]. In the present study, the A156S, A398V, and D399Y mutations in the N protein were detected at frequencies exceeding 3%.

The current study has some limitations. We only sequenced the complete genomes of SARS-CoV-2-positive specimens with high viral loads, which could be associated with specific SARS-CoV-2 genotypes. Secondly, the study focused solely on phylogeny and molecular characteristics; therefore, inferences about the antigenicity of new SARS-CoV-2 variants were limited. The genetic sequence data used in the study were not from samples that were randomly selected for sequencing; hence, they may not be representative of the SARS-CoV-2 circulating throughout Thailand. Lastly, the study did not investigate correlations between different SARS-CoV-2 variants and clinical features, thus missing an opportunity to identify potential changes in clinical manifestations associated with emerging SARS-CoV-2 variants.

In summary, the present study highlighted the changing SARS-CoV-2 variants in epidemic waves in Thailand and identified unique genomic patterns that may be associated with the severity of COVID-19. The occurrence of some mutations can significantly affect the evolutionary trajectory of the epidemic and the dissemination of genetically diverse variations. Continued molecular surveillance, including complete genome sequencing, is crucial with respect to identifying emerging SARS-CoV-2 variants early. This will enable us to reduce the overall burden of COVID-19 and guide research on SARS-CoV-2 vaccines and therapeutic targets.

## Figures and Tables

**Figure 1 viruses-15-01394-f001:**
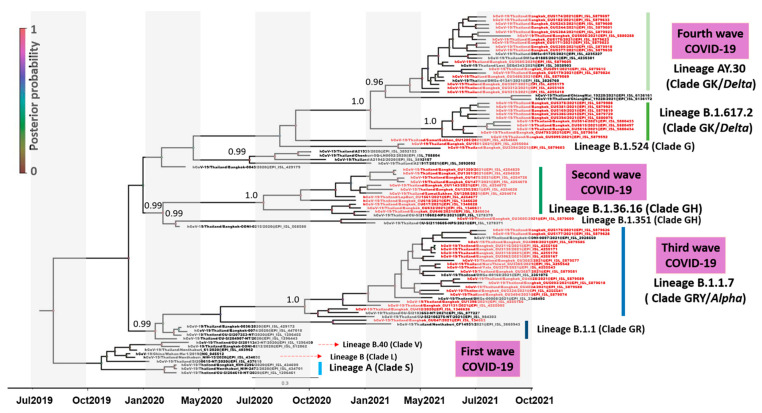
Time-scaled phylogenetic tree of 96 complete SARS-CoV-2 genomes (nt positions 56–29,739, 29,684 bp) detected before the B.1.1.529 omicron variant predominated. Shown is a maximum clade credibility tree constructed from 10,000 trees sampled from the posterior distribution with mean node ages. Clades described in GISAID are identified (S, L, V, G, GH, GR, GRY, and GK). Several lineages predominantly represent outbreaks in Thailand, and posterior probability support is given.

**Figure 2 viruses-15-01394-f002:**
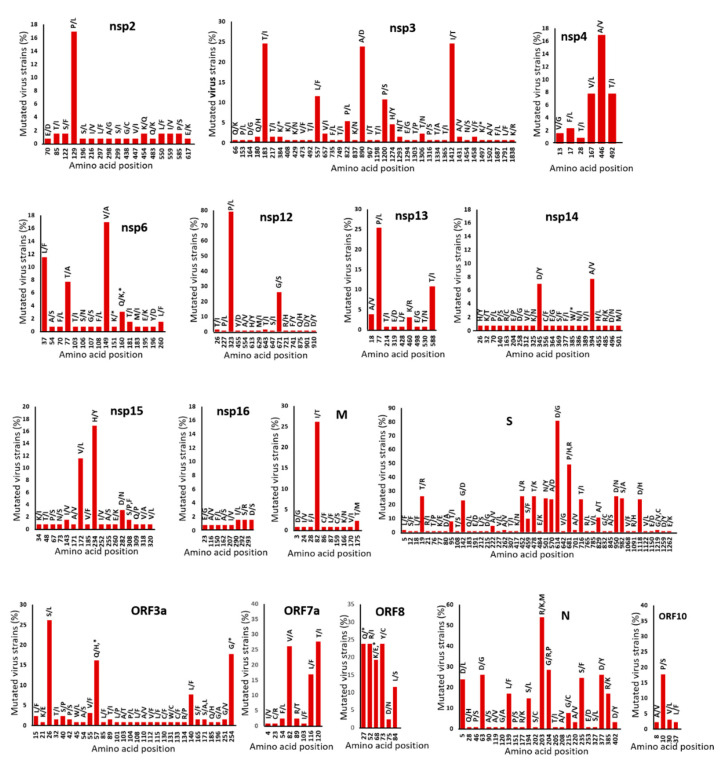
Amino acid mutations in the 130 SARS-CoV-2 genomes analyzed in the study (nt positions 56–29,739, 29,684 bp), compared to the Wuhan-Hu-1 (accession NC_045512) reference strain. The percentage frequency of all amino acid positions in the 130 genomes is shown on the *y*-axis. NSP, nonstructural protein; M, membrane protein; S, spike protein; N, nucleoprotein; ORF, open reading frame encoding the accessory protein.

**Figure 3 viruses-15-01394-f003:**
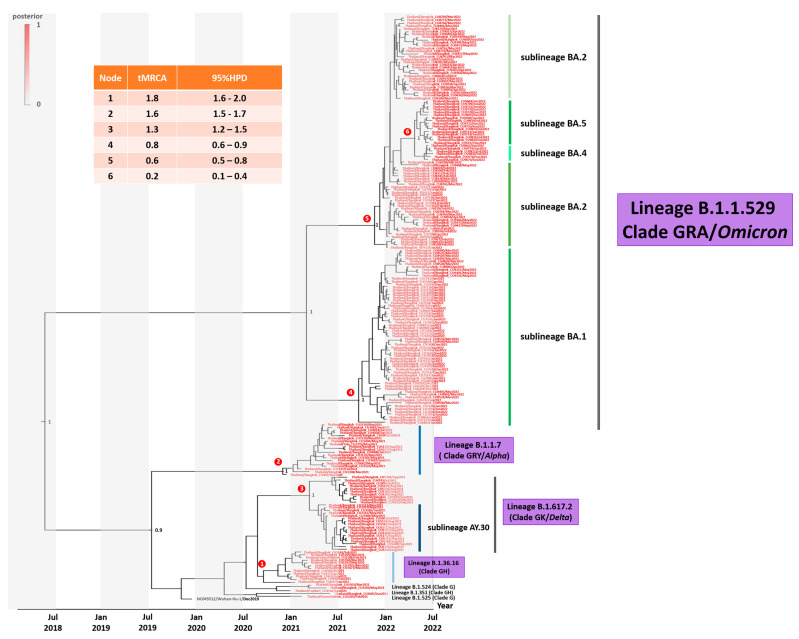
Time–scaled phylogenetic tree of complete spike sequences (nt positions 21,566–25,387, 3831 bp) of SARS-CoV-2 variants. Shown is a maximum clade credibility tree constructed from 10,000 trees sampled from the posterior distribution with mean node ages. Several lineages predominantly represent outbreaks in Thailand, and posterior probability support is given.

**Figure 4 viruses-15-01394-f004:**
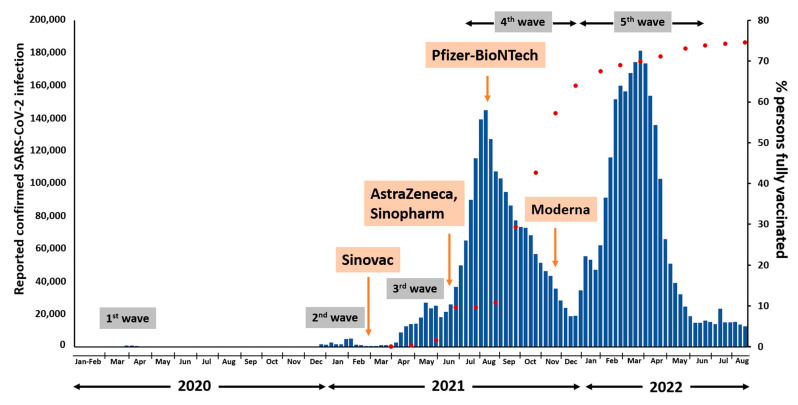
Timeline of the COVID-19 vaccination in Thailand.

**Figure 5 viruses-15-01394-f005:**
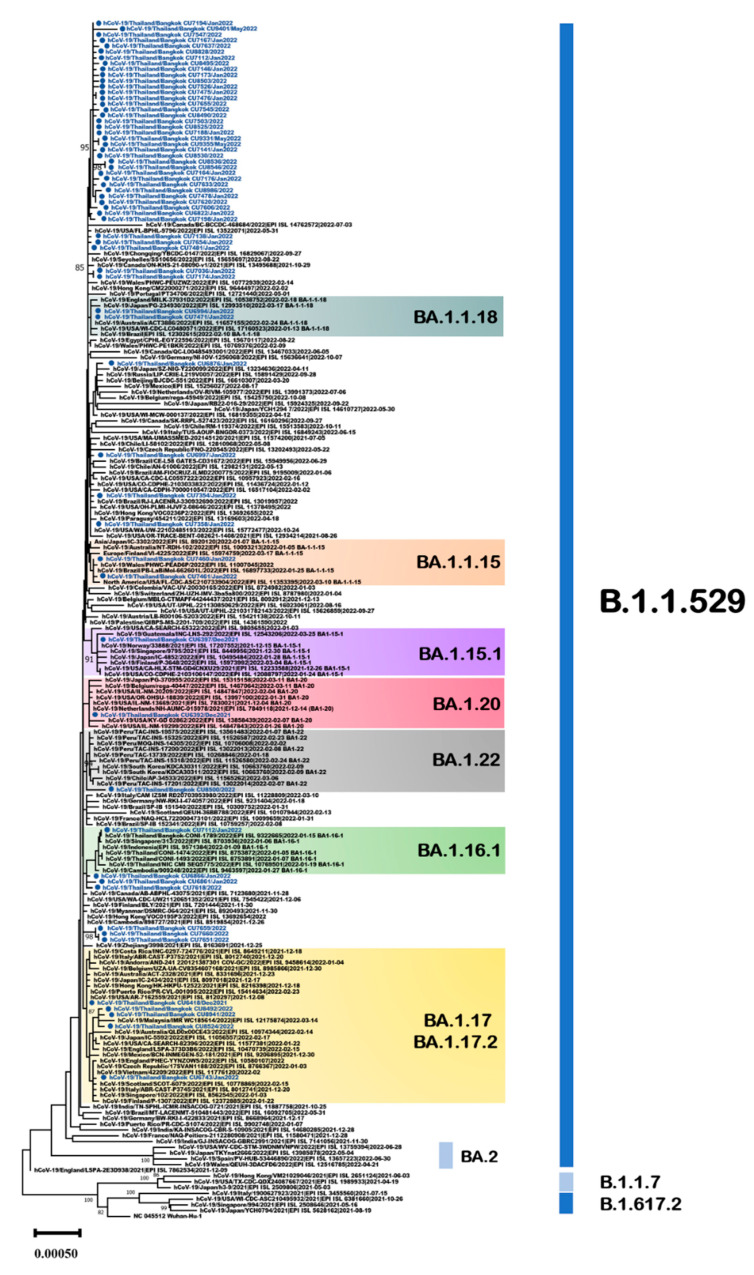
Unrooted phylogenetic analyses of SARS-CoV-2 omicron sublineage BA.1 variant based on full genome sequences (nt positions 202–29,745, 29,544 bp). Bootstrap values for key nodes are shown as percentages of 1000 replicates. All SARS-CoV-2 omicron sublineage BA.1 variants identified in this study are represented and labeled. Scale bars represent the number of substitutions per site.

**Figure 6 viruses-15-01394-f006:**
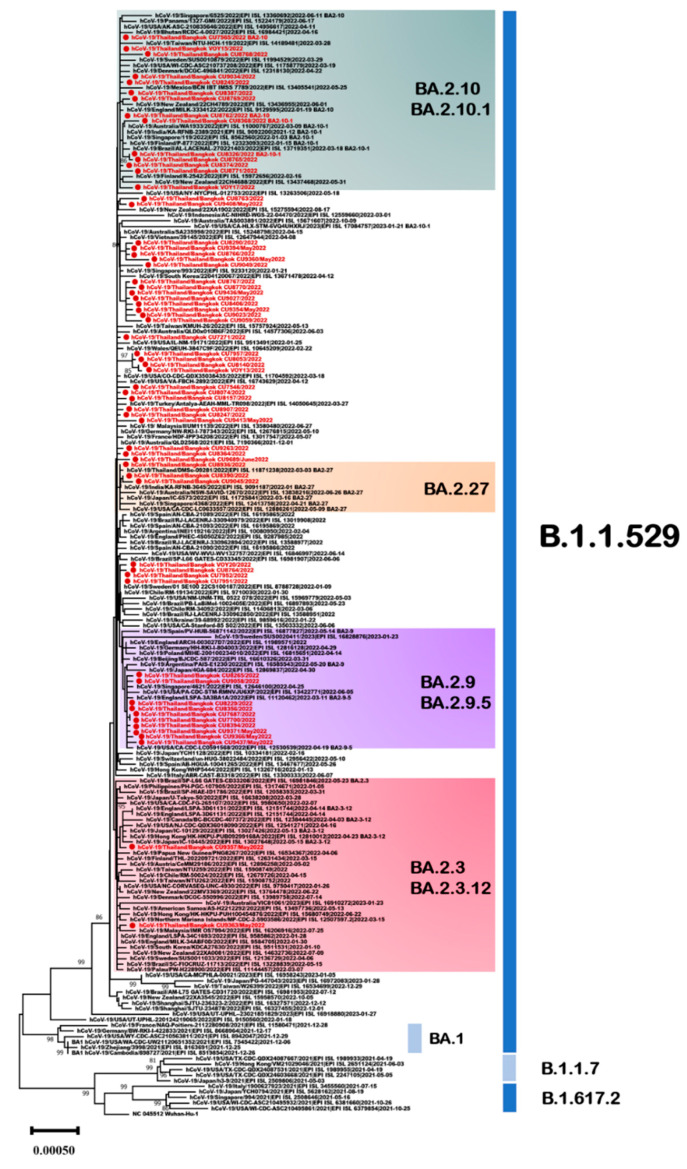
Unrooted phylogenetic analyses of SARS-CoV-2 omicron sublineage BA.2 variants based on full genome sequences (nt positions 218–29,686, 29,469 bp). Bootstrap values for key nodes are shown as percentages of 1000 replicates. All SARS-CoV-2 omicron sublineage BA.2 variants identified in this study are represented and labeled. Scale bars represent the number of substitutions per site.

**Figure 7 viruses-15-01394-f007:**
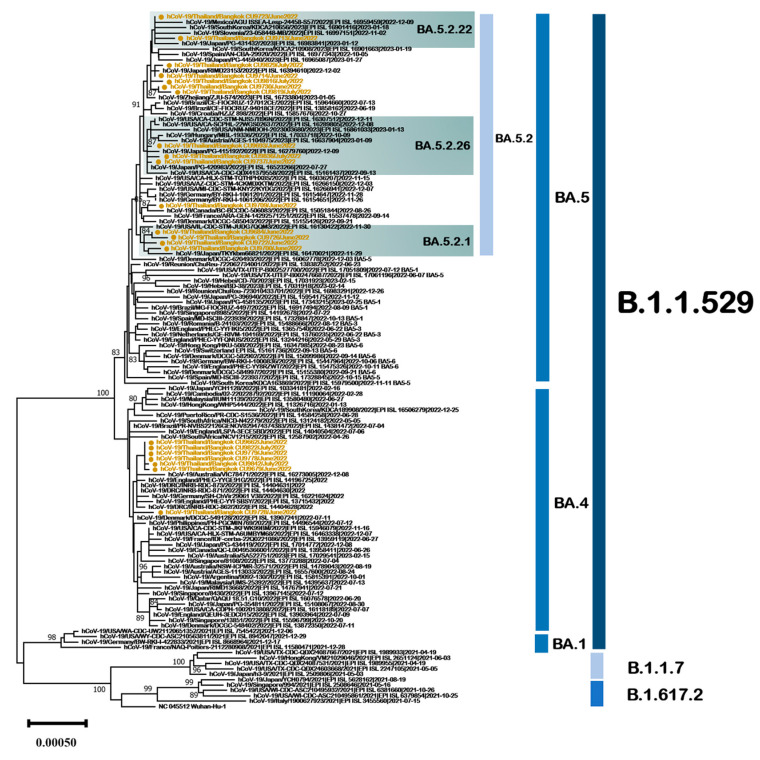
Unrooted phylogenetic analyses of SARS-CoV-2 omicron sublineages BA.4 and BA.5 variants based on full genome sequences (nt positions 201–29,698, 29,496 bp). Bootstrap values for key nodes are shown as percentages of 1000 replicates. All SARS-CoV-2 omicron BA.4 and BA.5 variants identified in this study are represented and labeled. Scale bars represent the number of substitutions per site.

**Table 1 viruses-15-01394-t001:** Comparison of missense and synonymous mutation frequency profiles of SARS-CoV-2 omicron sublineage BA.1 in Thailand datasets (nt length 29,544 bp).

Lineage	Gene	nt Position	aa Position	Genome Count	Frequency
BA.1	ORF1ab	T2019C	M585T	135	16.40
(*n* = 823)		C2470T		89	10.81
		G5515T		5	0.61
		G6850T		146	17.74
		C15952T		3	0.36
	S	G23628A	S689N	99	12.03
		C26936T		58	7.05
BA.1.1	ORF1ab	C2470T		1374	96.83
(*n* = 1419)		G3692A	V1143I	8	0.56
		G3896T	V1211F	21	1.48
		G6109A		75	5.29
		C11750T	L3829F	11	0.78
		G12661A		116	8.17
		C14805T		735	51.80
		G18433A	D1656N	35	2.47
		T19632C		744	52.43
	ORF3a	G25634A	C81Y	8	0.56
	E	G26428T	V62F	12	0.85
	M	A26530G	D3G	1060	74.70
	N	C28838T	R189C	29	2.04
BA.1.1.5	ORF1ab	C14117T	T217M	51	64.56
(*n* = 79)	S	C21597T	S12F	6	7.59
	M	A26530G	D3G	65	82.28
BA.1.1.8	ORF1ab	G2894A	D877N	10	13.70
(*n* = 73)	ORF3a	G26167T	V259L	9	12.33
BA.1.15.1	M	A26530G	D3G	12	46.15
(*n* = 26)					
BA.1.16.1	ORF1ab	G1806A	G514E	13	9.35
(*n* = 139)		C6401T	P2046S	18	12.95
	M	A26530G	D3G	127	91.37
	N	G29162A	D297N	4	2.88
		C29274T	T334I	4	2.88
BA.1.17	ORF1ab	T851C	Y196H	15	3.98
(*n* = 377)		C4113T	A1283V	176	46.68
		C5672T	P1803S	156	41.38
		C10605T	P3447L	12	3.18
		C12084T	T3940I	13	3.45
		G15850A	D795N	30	7.96
	M	A26530G	D3G	322	85.41
	N	G28436T	A55S	5	1.33
BA.1.20	ORF1ab	C15830T	A788V	4	26.67
(*n* = 15)	M	A26530G	D3G	5	33.30
BA.1.22	ORF1ab	G11083T	L3606F	7	7.87
(*n* = 89)		C15928T	P821S	6	6.74
	N	C29466T	A398V	7	7.87

**Table 2 viruses-15-01394-t002:** Comparison of missense and synonymous mutation frequency profiles of SARS-CoV-2 omicron sublineage BA.2 isolates in the Thailand dataset (nt length 29,469 bp).

Lineage	Gene	nt Position	aa Position	Genome Count	Frequency
BA.2	5′ UTR	C241T		1274	87.26
(*n* = 1460)	ORF1ab	C6196T		194	13.29
		C7471T		610	41.78
		C854T	P197S	26	1.78
		C3653T	L1130F	26	1.78
		C3686T	H1141Y	64	4.38
		C4893T	T1543I	32	2.19
		A4916G	I1551V	10	0.68
		C6401T	P2046S	45	3.08
		G7798T	K2511N	12	0.82
		C10789T		16	1.10
		C11109T	A3615V	26	1.78
		G14188A	A241T	42	2.88
		C15240T		194	13.29
		G15451A	G662S	13	0.89
		C16362T		88	6.03
		A19133C	E1889A	15	1.03
	ORF3a	A25411G	I7V	17	1.16
		C25613T	S74F	21	1.44
	S	C22120A	F186L	15	1.03
		G22632A	R357K	17	1.16
		T22882G	N440K	1270	86.99
		C23280T	T573I	49	3.36
		C23854A	N764K	1418	97.12
		T25224C	I1221T	25	1.71
		C25416T		582	39.86
	N	G29468T	D399Y	44	3.01
BA.2.27	5′ UTR	C241T		222	84.73
(*n* = 262)	ORF1ab	C10198T		242	92.37
		C12403T		58	22.14
		C17745T		20	7.63
		C19610T	T2048I	12	4.58
	ORF3a	C25672T	L94F	15	5.73
	N	G28739T	A156S	10	3.82
BA.2.3	5′ UTR	C241T		402	88.55
(*n* = 454)	ORF1ab	C832T		98	21.59
		T7282C		144	31.72
		C14267T	T267M	39	8.59
		C18508T	L1681F	37	8.15
		A21222G		358	78.85
BA.2.9	5′ UTR	C241T		451	86.90
(*n* = 519)	ORF1ab	G1820A	G519S	78	15.03
		A2442C	E726A	36	6.94
		T4443C	V1393A	87	16.76
		C5051T	P1596S	58	11.18
		C5672T	P1803S	42	8.09
		C12789T	T4175I	32	6.17
		A14109G	I214M	38	7.32
		A15553G	N696D	13	2.50
		T16494C		36	6.94
		C18457T	P1664S	11	2.12
	ORF9b	A28389T	N36Y	28	5.39
	S	T21752A	W64R	59	11.37
		T22882G	N440K	448	86.32
		G24348T	S929I	10	1.93
BA.2.10	5′ UTR	C241T		871	94.16
(*n* = 925)	ORF1ab	C2676T	P804L	15	1.62
		A4457G	I1398V	21	2.27
		T7813C		262	28.32
		C17528T	T1354I	71	7.68
	ORF3a	C25961T	T190I	120	12.97

**Table 3 viruses-15-01394-t003:** Comparison of missense and synonymous mutation frequency profiles of SARS-CoV-2 omicron sublineage BA.4 and BA.5 variants in the Thailand dataset (nt length 29,496 bp).

Lineage	Gene	nt Position	aa Position	Genome Count	Frequency
BA.4	5′ UTR	C241T		150	81.97
(*n* = 183)	ORF1ab	G6680A	A2139T	158	86.34
		T15521A	F685Y	9	4.92
	S	A22786C	R408S	154	83.70
		T22882G	N440K	134	73.22
		T24163C		160	87.43
BA.5.2	5′ UTR	C241T		1406	79.98
(*n* = 1758)	ORF1ab	C823T		140	7.96
		C5497T		712	40.50
		C13551T		82	4.66
		T16023C		716	40.73
		C16616A	T1050N	1708	97.16
		A18163G	I1566V	1617	91.98
	S	T22882G	N440K	1412	80.32
		C23854A	N764K	1596	90.78
	E	C26270T	T9I	1342	76.34

## Data Availability

Genome sequences generated in this study were deposited in the GISAID (https://www.gisaid.org, accessed on 7 January 2023) databases. Accession IDs are available in Appendix A.

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
