# Peer review of "Investigation of the Molecular Epidemiology and Evolution of Circulating Severe Acute Respiratory Syndrome Coronavirus 2 in Thailand from 2020 to 2022 via Next-Generation Sequencing"

_viruses, 2023, doi:10.3390/v15061394_

Round 1
Reviewer 1 Report
The paper describes the molecular epidemiology and evolution of CoVID-19 in Thailand from 2020 and 2022. A WGS of 210 samples collected were processed for NGS using Illumina chemistry. Of these 210 sequences, 63 were detected before the omicron variant and remaining 147 sequences were collected during the omicron variant predominance.
The introduction, review and literature, MM, results and discussion are well written. Minor corrections and some clarifications are needed which is as under please:
1. Line 133: MCMC needs to be elaborated in words.
2. Line 193-194: The S, N and M SP changes in amino acid 46, 24 and 10 is not shown in the figure 2.
3. Line 307: Please clarify where?
4: Fig 6 : Does not show subtypes 5.2.1, 22 and 26 please clarify.'
The Institution review board approval has been taken and well explained.
Author Response
Response to Reviewer 1 Comments
The paper describes the molecular epidemiology and evolution of CoVID-19 in Thailand from 2020 and 2022. A WGS of 210 samples collected were processed for NGS using Illumina chemistry. Of these 210 sequences, 63 were detected before the omicron variant and remaining 147 sequences were collected during the omicron variant predominance.
The introduction, review and literature, MM, results and discussion are well written. Minor corrections and some clarifications are needed which is as under please:
- Line 133: MCMC needs to be elaborated in words.
Response 1: This has been revised (line 133).
- Line 193-194: The S, N and M SP changes in amino acid 46, 24 and 10 is not shown in the figure 2.
Response 2: This sentence has been revised in line 193-194.
- Line 307: Please clarify where?
Response 3: This has been clarified (line 307).
4: Fig 6 : Does not show subtypes 5.2.1, 22 and 26 please clarify.'
Response 4: The figure 6 has been changed.

Reviewer 2 Report
Nicely presented study using conventional methods
- not particularly novel, though the findings will be of interest to those looking at how SARS-COV-2 spread in Thailand
- please just clarify in the BEAST analysis - the number of gamma categories used, and the discarded burn-in (10% of 120 million with BEAST - then 50 million when examining the output with TRACER?)
- for each Fig/Table for each variant (BA1, BA2, BA.4/BA.5),please include the exact final length of the sequences used in the tree construction/mutation analysis - you always lose some cases during the final alignment
- finally, can the authors add a timeline of the COVID-19 vaccination in Thailand and which vaccines were used, and what proportion of the population were vaccinated at each point - and discuss how the COVID vaccines could have affected what they saw in their analysis - and it’s interpretation?
Author Response
Response to Reviewer 2 Comments
Nicely presented study using conventional methods
- not particularly novel, though the findings will be of interest to those looking at how SARS-COV-2 spread in Thailand
Response 1: The sample sizes in this study are not sufficient for studying how SARS-CoV-2 spreads in Thailand, and the samples were detected from only one province. The author has mentioned this limitation in the study.
- please just clarify in the BEAST analysis - the number of gamma categories used, and the discarded burn-in (10% of 120 million with BEAST - then 50 million when examining the output with TRACER?)
Response 2: This has been revised in the BEAST analysis (line 134-142).
- for each Fig/Table for each variant (BA1, BA2, BA.4/BA.5),please include the exact final length of the sequences used in the tree construction/mutation analysis - you always lose some cases during the final alignment
Response 3: These have been added in all Fig/Table legends.
- finally, can the authors add a timeline of the COVID-19 vaccination in Thailand and which vaccines were used, and what proportion of the population were vaccinated at each point - and discuss how the COVID vaccines could have affected what they saw in their analysis - and it’s interpretation?
Response 4: These have been added in Figure 4 and line 237-247 and 372-373.
